# Lignin oxidation products in soil, dripwater and speleothems from four different sites in New Zealand

Inken Heidke[1], Adam Hartland[2], Denis Scholz[3], Andrew Pearson[2], John Hellstrom[4], Sebastian F. M. Breitenbach[5], and Thorsten Hoffmann[1]

[1]Institute of Inorganic Chemistry and Analytical Chemistry, Johannes Gutenberg-University of Mainz, Duesbergweg 10-14, 55128 Mainz, Germany
[2]Environmental Research Institute, School of Science, University of Waikato, Private Bag 3105, Hamilton, 3240, New Zealand
[3]Institute for Geosciences, Johannes Gutenberg-University of Mainz, J.-J.-Becher-Weg 21, 55128 Mainz, Germany
[4]School of Earth Sciences, University of Melbourne, 253-283 Elgin St, Carlton VIC 3053, Australia
[5]Department of Geography and Environmental Sciences, Northumbria University, Newcastle upon Tyne, NE1 8ST, United Kingdom

**Correspondence:** Thorsten Hoffmann (t.hoffmann@uni-mainz.de)

**Abstract.** Lignin oxidation products (LOPs) are widely used as vegetation proxies in climate archives, such as sediment and peat cores. The total LOP concentration, $\Sigma 8$, provides information on the abundance of vegetation, while the ratios C/V and S/V of the different LOP groups also provide information on the type of vegetation. Recently, LOP analysis has been successfully applied to speleothem archives. However, there are many open questions concerning the transport and microbial degradation of LOPs on their way from the soil into the cave system. These processes could potentially alter the original source-dependent LOP signals, in particular the C/V and S/V ratios, and thus complicate their interpretation in terms of past vegetation changes. We analyzed LOPs in leaf litter and different soil horizons as well as dripwater and flowstone samples from four different cave sites from different vegetation zones in New Zealand using ultrahigh performance liquid chromatography coupled to high resolution mass spectrometry. We test whether the original source-dependent LOP signal of the overlying vegetation is preserved and can be recovered from flowstone samples and investigate how the signal is altered by the transport from the soil to the cave. The LOP concentrations range from mg/g in the soil to ng/g in the flowstones. Our results demonstrate that, from the soil to the flowstone, the C/V and S/V ratios both increase, while the total lignin content, $\Sigma 8$, strongly decreases. This shows that the LOP signal is strongly influenced by both transport and degradation processes. Nevertheless, the relative LOP signal from the overlying soil at the different cave sites is preserved in the flowstone. We emphasize that for the interpretation of C/V and S/V ratios in terms of past vegetation changes, it is important to compare only samples of the same type (e.g., speleothem, dripwater or soil) and to evaluate only relative variations.

# 1 Introduction

Climate archives provide the means to study the climate and environment of the past, which is necessary to put ongoing climate change into a historic framework. Speleothems are valuable climate archives because they are ubiquitous across climatic and vegetational zones, preserve a range of inorganic and organic proxies (Fairchild and Baker, 2012; Blyth et al., 2016) and can

be accurately dated up to ca. 600,000–700,000 years before present using U-series dating (Cheng et al., 2016; Scholz and Hoffmann, 2008). While up to the 1990s, speleothem research focused mainly on stable isotopes (e.g., $\delta^{18}$O, $\delta^{13}$C) (McDermott, 2004), today, multi-proxy studies combining, for example, stable isotopes, trace elements, fluorescent organic matter and selected molecular organic biomarkers are employed to unravel climatic and ecological signals (Blyth et al., 2016; Fairchild et al., 2006). Vegetation proxies not only complement other climate proxies, but inform about the impact of changing temper-

ature and precipitation on the development of vegetation. Fatty acids, originating from both plants and soil microorganisms, have been extracted from speleothems to study soil and vegetation activity (Blyth et al., 2006; Bosle et al., 2014), and long-chain *n*-alkanes from plant leaf waxes have been used to investigate vegetation changes (Xie, 2003; Blyth et al., 2007, 2011). However, the use of *n*-alkane chain length distributions to distinguish between different plant groups has been debated (Bush and McInerney, 2013; Blyth et al., 2016).

Lignin is a promising paleo-vegetation proxy because it is produced exclusively by vascular plants and represents one of the main constituents of wood and woody plants (Boerjan et al., 2003; Jex et al., 2014). In addition, the analysis of lignin has the particular advantage that it can yield information not only about the abundance, but also the type of vegetation. Lignin is a biopolymer that mainly consists of three different monomers: coniferyl alcohol, sinapyl alcohol, and p-coumaryl alcohol, which are linked via C-C and C-O bonds in a radical coupling mechanism. The resulting structural units are guaiacyl (G),

syringyl (S) and p-hydroxyphenyl (H) phenylpropanoid units, respectively, with the corresponding ratios varying with plant type. Lignin from gymnosperms (soft wood) consists mainly of G units, whereas lignin from angiosperms (hard wood) contains both G and S units. Grasses and non-woody plant parts, such as leaves and needles, are constructed from equal parts of G, S and H units. In addition, p-coumaric acid and ferulic acid can be ester-bound to the terminal hydroxyl-groups of the propyl side chains, especially in grass lignin (Boerjan et al., 2003; Kögel-Knabner, 2002).

To analyze the lignin composition, the lignin polymer is oxidatively degraded into monomeric lignin oxidation products (LOPs), for example via CuO oxidation. The guaiacyl unit is oxidized to vanillic acid, vanillin and acetovanillone (V-group LOPs), and the syringyl unit to syringic acid, syringaldehyde and acetosyringone (S-group LOPs). Ferulic acid and p-coumaric acid form the C-group LOPs. The p-hydroxyphenyl unit is oxidized to p-hydroxybenzoic acid, p-hydroxybenzaldehyde and p-hydroxyacetophenone (H-group LOPs), but because these can also originate from other sources than lignin, such as soil

microorganisms or the degradation of protein-rich material, the H-group LOPs are not used as vegetation proxies. The sum of all eight individual LOPs from the C-, S- and V-group, $\Sigma 8$, is used to represent the total lignin concentration, whereas the ratios of the different LOP groups, C/V and S/V, are used to represent the type of lignin, with a higher C/V ratio indicating a higher contribution of non-woody vs. woody plant material and a higher S/V ratio indicating a higher contribution of angiosperm vs. gymnosperm plant material (Hedges and Mann, 1979).

The microbial degradation of lignin in the soil is comparably slow. For instance, it can take several years to break down a piece of wood (Opsahl and Benner, 1995). This is because only white-rot fungi are able to completely mineralize lignin to $CO_2$, mostly by co-metabolism with other energy sources. Other fungi, such as soft-rot and brown-rot fungi, are only able to induce structural changes to lignin (Kögel-Knabner, 2002). Due to this relative stability, lignin oxidation products are widely
used as a paleo-vegetation proxy in climate archives, such as peat, lake sediments and marine sediment cores (see the review by Jex et al., 2014). In addition, LOPs are used as a proxy for the terrestrial input of plant biomass in natural waters (Zhang et al., 2013; Standley and Kaplan, 1998; Hernes and Benner, 2002). Blyth and Watson (2009) first detected lignin-derived compounds in speleothems, and lignin was subsequently highlighted as a promising vegetation proxy in speleothems in several review papers (Blyth et al., 2008; Jex et al., 2014; Blyth et al., 2016).

A recent study showed that trace concentrations of lignin oxidation products can be quantified in dripwater and speleothems (Heidke et al., 2018). In addition, in a Holocene stalagmite from the Herbstlabyrinth, Germany, the LOP concentrations and C/V and S/V ratios were found to co-vary with environmental changes and correlate with other proxies, such as P, U and Ba concentrations as well as $\delta^{13}C$ values (Heidke et al., 2019). This study also observed seasonal variations in LOP concentrations in monthly dripwater samples from the Herbstlabyrinth, with lower LOP concentrations in winter and higher concentrations in
summer. However, there are still many open questions concerning the transport of lignin from the soil through the karst system into the cave and speleothem. For instance, the interaction of lignin with mineral surfaces can lead to fractionation (Hernes et al., 2007, 2013), and land-use, climate, and soil characteristics may also influence the degradation of lignin (Thevenot et al., 2010; Jex et al., 2014). These effects could potentially alter the original source-dependent LOP signals, in particular the C/V and S/V ratios, and thus complicate their interpretation in terms of past vegetation changes. A schematic overview of the
potential processes influencing the LOP signals is shown in Fig. 1.

The vast majority of procedures for the determination of organic analytes in aqueous solutions such as drip water rely on the collection of discrete samples of the water at a specific time. Subsequent laboratory analysis of the sample then provides a snapshot of the concentration of the target analytes at the time of sampling. In the presence of fluctuating concentrations, this method also has drawbacks, such as allowing episodic concentration fluctuations to be missed. One solution to this problem is
to increase the frequency of sampling or to install automated sampling systems that can collect numerous drip water samples over a period of time. However, this is costly and in many cases impractical. But alternatives exist to overcome some of these difficulties. Of these, passive sampling methods have shown to be promising tools for measuring aqueous concentrations of a wide range of organic substances. Passive samplers avoid many of the problems described above because they enrich target analytes in situ, can be used for extended periods of time, and can be applied without continuously accessing the sampled
caves. The goal is to determine the mass of target components accumulated by a sampler, thereby obtaining time-averaged concentrations. Of course, passive samplers also have their disadvantages (Vrana et al., 2005), which is why they are used here as one method in combination with other objects of investigation (leaf litter, soil as well as flowstone samples) and are primarily used to compare the different sampled sites with each other.

In this study, we analyzed the LOP signals in soil and dripwater samples as well as recently deposited flowstones from six
different cave sites in New Zealand with different overlying vegetation. We investigated whether the original source-dependent

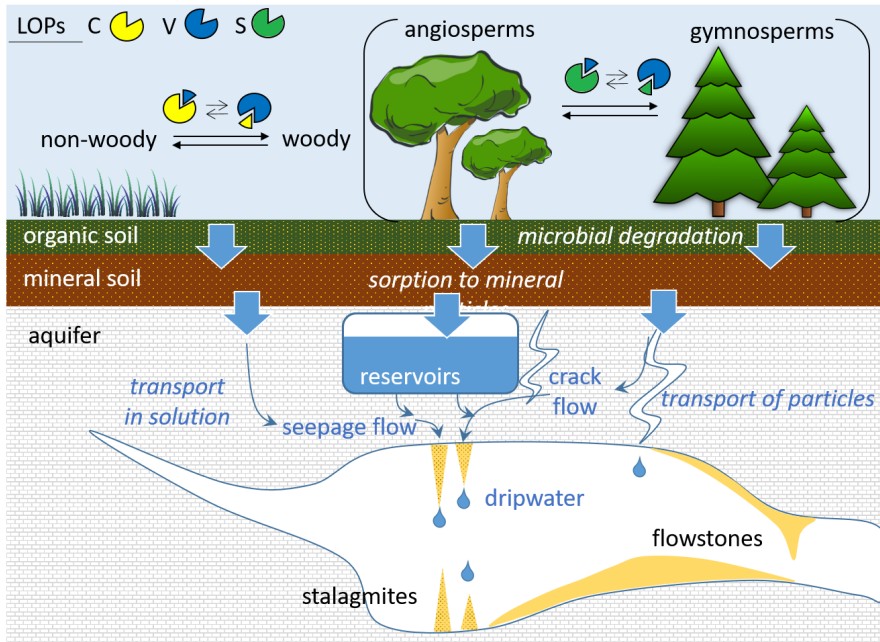

**Figure 1.** Schematic representation of the different potential processes influencing the transport and degradation of LOPs in speleothems and cave dripwater.

signal of the overlaying vegetation is preserved and can be recovered from the flowstone samples. Furthermore, we investigated how the signal is altered by the transport process.

## 2   Methods and materials

### 2.1   Location and environment of the cave sites

5   The samples in this study were taken from six different cave sites in New Zealand (Fig. 2), spanning a latitudinal gradient from 38°S (Waipuna Cave) to 45°S (Luxmore Caves). Waipuna Cave (WP) is situated in the Waitomo district in the west of the North Island (S 38.3114722°, E 175.0206389°). The vegetation above the cave consists of pasture and mixed podocarp-broadleaved forest dominated by tree ferns. Soils are developed from rhyolitic tephra and are characterized by very deep A-horizons. Hodges Creek Cave (HC, S 41.171270°, E 172.685941°) and Nettlebed Cave (NB, S 41.2104589°, E 172.7394572°) are both situated

10   in Kahurangi National Park, 40° S, in the Tasman district in the north-west of the South Island. The vegetation above both caves consists mainly of mature southern beech forest with well developed O-horizons. Luxmore Cave (LX), Daves Cave (DC) and Calcite Cave (CC) are all situated on Mt. Luxmore in Fiordland National Park in the Southland district in the south-west of the South Island (S 45.3894900°, E 167.6153448°). The vegetation here consists mainly of tussock grassland with alpine species present. A description of all sites including photographs of the vegetation and the soils is provided in the supplement.

From all caves, flowstones were selected for their higher organic matter content and their tendency to capture more diverse flow paths from surface environments compared to other speleothems such as stalagmites.

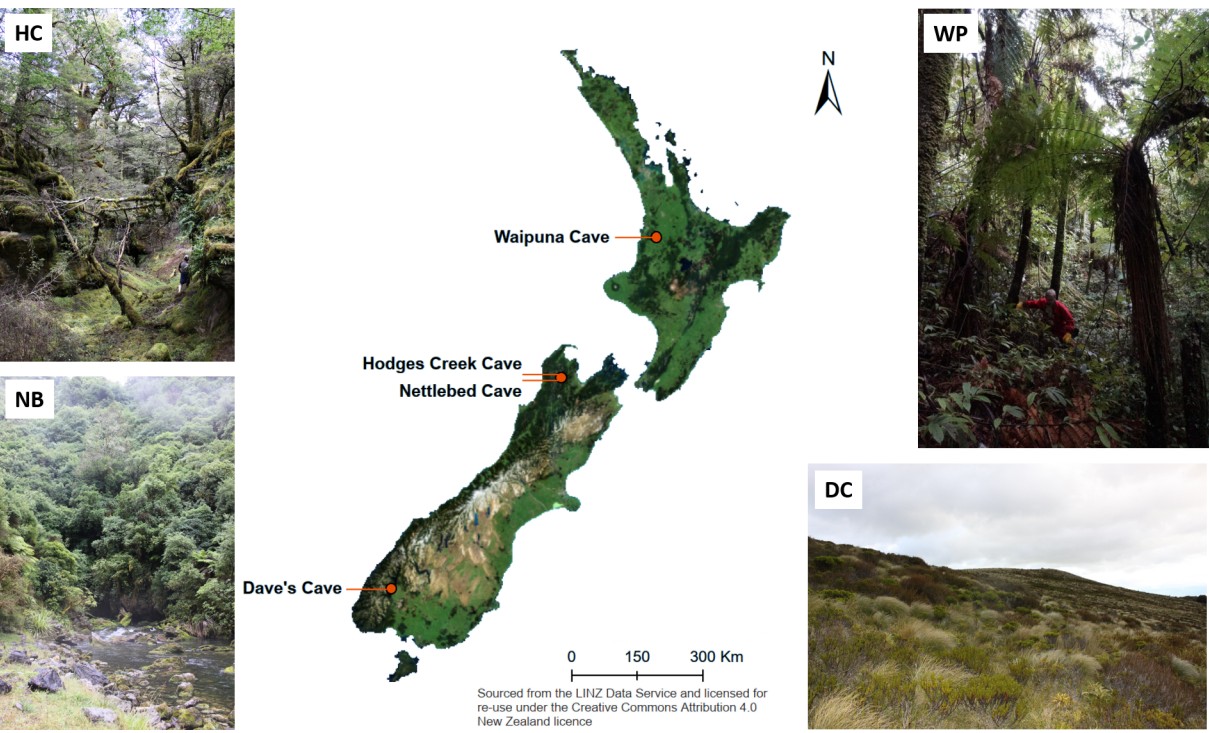

**Figure 2.** Map of New Zealand with the locations of the caves indicated. The photos give an impression of the vegetation at the sites of Hodges Creek Cave (HC), Waipuna Cave (WP), Nettlebed Cave (NB) and Dave's Cave (DC).

## 2.2 Sampling and pretreatment of samples

The overall procedure for the analysis of lignin oxidation products (LOPs) was similar to the procedure described in Heidke et al. (2018). However, since different types of samples were used, the methodology differed in some points and is thus briefly described in the following paragraphs.

### 2.2.1 Sampling and pretreatment of soil and litter samples

Soil and litter samples were collected during field trips in 2014–2016 and kept refrigerated at 4°C in sealed containers prior to analysis. Soil horizons (A and O) and leaf litter (LL) were assigned on the basis of depth, colour and composition of the soil with reference to the New Zealand soil classification manual (Hewitt, 2010) at the time of field sampling. Later, a subsample was removed, sieved over a 2.75 mm sieve and dried at room temperature beneath a fume hood. The dried samples were finely ground with a pestle and mortar and weighed into microwave reaction vessels. All soil samples were prepared and analyzed in duplicate. 0.5 g aliquots were taken from samples WP-LL, NB-LL and HC-LL (LL = leaf litter), 0.75 g were taken from

NB-O (O = O-horizon), and 1.0 g were taken from all other samples (NB-A (A = A-horizon), LX-LL, LX-O, LX-A, WP-O, WP-A, HC-O and HC-A). 500 mg of CuO, 100 mg of $(NH_4)_2Fe(SO_4)$ and 15 mL of $2 \, mol \cdot L^{-1}$ NaOH were added into the microwave reaction vessels, and microwave-assisted degradation was performed as described in section 2.3

### 2.2.2 Active sampling and pretreatment of dripwater samples

Dripwater from four different drip sites in Waipuna Cave was sampled in LDPE bottles that were precleaned with double-distilled water. The drip sites are well studied for their dynamics by Nava-Fernandez et al. (2020). Drip site WP2 is a highly seasonal drip site exhibiting high discharge following rainfall events. 1.0 L samples were collected from WP2 in April, July, September and November 2017 to investigate seasonal changes. From all other drip sites, 0.5 L samples were collected in November 2017. Drip site WP3 is a fast drip site with 0.5 L of water collected in about one hour, WP4 is a slow drip site with

0.5 L of water collected within four days, and WP5 was a very fast drip site with 0.5 L of water collected within a few minutes. In addition, 1.0 L of water from a stream flowing through the cave was sampled in November 2017. After collection, the samples were transferred into precleaned glass bottles and 5% (w/w) of acetonitrile was added to the water to prevent microbial growth. The samples were stored in a fridge at 4 °C awaiting further processing. Before extraction via solid phase extraction (SPE), the samples were acidified to pH 1–2 with conc. HCl and filtered through 0.45 µm cellulose acetate membrane filters. WP2 water

samples and the stream water were divided into three subsamples equating to 0.33 L each, all other samples were divided into two subsamples equating to 0.25 L each. The SPE cartridges (Oasis HLB, 200 mg sorbent, 6 mL volume) were preconditioned with 6 mL of MeOH followed by 6 mL of water acidified with HCl. The samples were loaded by gravity flow using sample reservoirs. Afterwards, the cartridges were rinsed twice with 6 mL of acidified water and dried for 20 min by sucking ambient air through the cartridges. The dried cartridges were stored in sealed plastic bags in the fridge for several weeks to months.

Eventually, the cartridges were eluted with 5 mL of MeOH in ten portions of 0.5 mL. The eluent was evaporated in a stream of nitrogen at 30 °C. The residue was redissolved in 1.5 mL of $2 \, mol \cdot L^{-1}$ NaOH and transferred into the microwave reaction vessel. 250 mg of CuO, 50 mg of $(NH_4)_2Fe(SO_4)$ and a total volume of 8 mL of $2 \, mol \cdot L^{-1}$ NaOH were added, and the microwave-assisted degradation was performed as described in section 2.3.

   The 0.45 µm cellulose acetate syringe filters that were used to filter the dripwater samples were stored in sealed plastic bags

in the fridge for several months. The number of filters used per sample varied depending on the particulate matter load. For sample WP2-04, 3 filters per subsample were used, adding up to 9 filters in total, and for sample WP2-09, four filters per subsample were used (12 filters in total). For all other samples, one filter per subsample was sufficient (2 filters in total for WP3, WP4 and WP5, 3 filters in total for WP2-07, WP2-11 and WP-stream). Three blank filters were stored, handled and analyzed in the same way as the sample filters. The filters were eluted with four portions of 1 mL of MeOH per filter. The

eluent was evaporated in a stream of nitrogen at 30 °C. The residue was dissolved in 3 mL of $2 \, mol \cdot L^{-1}$ NaOH and then transferred into the microwave reaction vessels for further degradation and analysis.

### 2.2.3 Passive sampling and pretreatment of dripwater samples

Passive sampling cartridges with 20 mL volume filled with XAD-7 polymeric resin were placed in the caves HC, WP, CC and DC underneath individual fast dripping drip sites for about one year in order to provide an integrated measure of dripwater organic matter fluxes. Cartridges were positioned so that the dripwater could flow through the cartridge, allowing organic

material to be adsorbed on the resin. Afterwards, the cartridges were capped and stored in the fridge before further processing. Eventually, the cartridges were rinsed with 10 mL deionized water and dried under vacuum. To extract the organic material, the XAD-7 resin was transferred into an Erlenmeyer flask and stirred for at least 30 min with 10 mL of an elution solvent mixture consisting of 95% MeOH, 5% $H_2O$ and $0.1 \; mol \cdot L^{-1}$ NaOH. The solvent mixture with the resin was then transferred back into the cartridge and the cartridge was eluted using positive pressure applied through a stream of nitrogen. The eluate was

collected and the resin extracted again with fresh solvent mixture. The procedure was repeated three times in total. Finally, the cartridges were eluted with 10 mL of the solvent mixture directly in the cartridges using positive pressure applied through a stream of nitrogen. The combined eluates were filtered over 1 μm glass fiber filters (Whatman GF/F) and evaporated at 35 °C in a stream of nitrogen, until only approx. 2 mL were left. This concentrated eluate, which consisted mainly of aqueous NaOH as solvent, was transferred into the microwave reaction vessel. 250 mg of CuO and 50 mg of $(NH_4)_2Fe(SO_4)$ were added, and

the volume was made up to 8 mL with $2 \; mol \cdot L^{-1}$ NaOH. The microwave-assisted degradation was performed as described in section 2.3.

### 2.2.4 $^{230}$Th/U-dating of flowstone cores

Flowstone cores from the study sites were sampled in 2015. Powdered speleothem samples of 100 to 200 mg were collected from sectioned core samples and analyzed as described in Hellstrom (2003) and Drysdale et al. (2012). Briefly, samples were

dissolved in conc. HNO3, spiked with a mixed $^{236}$U-$^{233}$U-$^{229}$Th isotopic tracer solution and capped and equilibrated on a hotplate overnight. U and Th were separated from the sample matrix using established procedure with Eichrom TRU ion exchange resin columns, eluted together into a teflon vial and dried down overnight. Sample U and Th were re-dissolved in dilute HNO3 and HF and introduced together via a teflon flow path to a Nu Instruments Plasma MC-ICP-MS via a Nu DSN desolvator. $^{238}$U ion beam intensities were typically 20 to 5 pA, with $^{233}$U/$^{234}$U and $^{229}$Th/$^{230}$Th analyzed simultaneously

in twin ion counters in a peak-jumping routine. $^{230}$Th/$^{238}$U and $^{234}$U/$^{238}$U activity ratios for samples were normalized to those calculated for uraninite equilibrium standards run in the same analytical session. Ages were calculated using equation 1 of Hellstrom (2006) and the decay constants of Cheng et al. (2013), with initial $^{230}$Th/$^{232}$Th and its uncertainty for each speleothem estimated using stratigraphic constraint. A table with all relevant data concerning the $^{230}$Th/U-dating as well the age-depths models can be found in the supplement.

### 2.2.5 Sampling and pretreatment of flowstone samples

Samples with 5–10 mm edge length were cut from the flowstone cores with a diamond blade saw following the direction of the growth layers. Photos of the flowstone cores are shown in the supplement. The distance from top of the flowstone core as well

as the ages of the LOP samples are given in Table 1. The flowstone cores were embedded in a polymeric resin for stabilization, which was not resistant to organic solvents, but could not be completely dissolved either. Therefore, the cleaning protocol described in Heidke et al. (2018) had to be slightly modified. Instead of ultrasonication with dichloromethane and methanol, the samples were merely rinsed first with EtOH, then with MeOH, and then dried in an air stream. The outer layer of calcite, where it was not covered by the resin, was etched in 0.6% HCl for 5 min. Afterwards, the samples were dried, weighed and then dissolved in 30% HCl. The sample weight was determined by weighing the resin after complete dissolution of the sample. The sample solution was diluted 1:1 with ultrapure water and then extracted via SPE as described in Heidke et al. (2018) using Oasis HLB 60 mg cartridges and eluted with 1.5 mL of methanol. The microwave-assisted degradation with 250 mg of CuO, 50 mg of $(NH_4)_2Fe(SO_4)$ and 8 mL of NaOH was performed as described in section 2.3.

**Table 1.** Distance from top (dft) of upper and lower edges of the flowstone sample cubes (dft(upper), (dft(lower)), ages calculated from the age model for upper and lower edges (age(upper), (age(lower)), and calculated mean sample ages for the flowstone samples (age(mean)).

| sample | dft(upper) in mm | dft(lower) in mm | age(upper) in ka BP | age(lower) in ka BP | age(mean) in ka BP |
|---|---|---|---|---|---|
| HC1 | 0 | 5 | 0.00 | 2.19 | $1.1 \pm 1.1$ |
| HC2 | 5 | 9 | 2.19 | 3.26 | $2.7 \pm 0.5$ |
| NB1 | 0 | 10 | 0.00 | 0.94 | $0.5 \pm 0.5$ |
| NB2 | 10 | 20 | 0.94 | 1.59 | $1.3 \pm 0.3$ |
| WP1 | 1 | 10 | 0.22 | 2.08 | $1.1 \pm 0.9$ |
| WP2 | 10 | 20 | 2.08 | 3.90 | $3.0 \pm 0.9$ |
| DC1 | 0 | 10 | 0.00 | 0.37 | $0.2 \pm 0.2$ |
| DC2 | 10 | 18 | 0.37 | 0.72 | $0.5 \pm 0.2$ |

## 2.3 CuO oxidation, extraction and analysis of lignin oxidation products

The reaction vessels filled with reagents and sample extracts were flushed with argon and capped. The oxidation, extraction and analysis procedures are described in detail in Heidke et al. (2018). In brief, the oxidation was carried out in a microwave oven at 155 °C for 90 min. After cooling, ethyl vanillin was added as an internal standard. The reaction mixture was separated from solids by centrifugation, acidified and then extracted via SPE. For the dripwater and flowstone samples, the whole sample solution was extracted, whereas for the soil samples, only 1 mL of the sample solution was extracted. Oasis HLB cartridges with 60 mg sorbent and 3 mL volume were used, which were eluted with 1 mL of acetonitrile containing 2% of $NH_4OH$.

The LC-MS analysis was performed similarly as described in Heidke et al. (2018). In brief, a pentafluorophenyl column with an acetonitrile-water gradient elution was used in an ultrahigh-performance liquid chromatography (UHPLC) system to separate the LOPs. The mass spectrometric detection was performed using a Q Exactive Orbitrap high-resolution mass

spectrometer, which was operated alternately in full-scan mode and MS/MS mode. Further details as well as slight differences to the method described by Heidke et al. (2018) can be found in the supplement.

## 3 Results and discussion

### 3.1 Comparison of soil, passively-sampled dripwater and flowstone samples from different cave sites

Comparing the LOP concentrations in soil, passively-sampled dripwater and flowstone samples as well as the distribution of the C-, S- and V-group LOPs (Fig. 3), the concentrations range from mg/g in the leaf litter and soil samples to ng/g in the flowstone samples. For the dripwater, the values are given in ng/sample, where 'sample' refers to an XAD-cartridge that was deployed in the cave for passive sampling for about one year. The total LOP concentrations, $\Sigma 8$, decrease strongly with soil depth in all caves (Fig. 3(a)). Large variations are observed between the different drip sites within the same cave, especially in

HC (Fig. 3(b)). The differences between the stratigraphically younger (1) and older (2) flowstone samples are less pronounced, with slightly higher LOP concentrations in the older samples, except for LX/DC cave (Fig. 3(c)). In all three samples types – soil, drip water and flowstones – the relation between the different cave sites regarding the total LOP concentration, $\Sigma 8$, is similar, with the highest LOP concentrations in samples from HC and NB and lower concentrations in samples from WP and LX (Fig. 3(e)).

In a scatter plot of S/V versus C/V values of soil, dripwater and flowstone samples (Fig. 4), each sample type occupies a distinct region of the diagram, with the soil samples having the lowest C/V values and low S/V values, the dripwater samples having medium C/V and the highest S/V values and the flowstone samples having the highest C/V and low to medium S/V values.

The C/V and S/V ratios in the leaf litter samples are representative of the current vegetation cover (Fig. 5), with low C/V

and high S/V ratios for HC and NB (angiosperm southern beech forest), low C/V and low S/V ratios for WP (gymnosperm podocarp forest, apparently dominating the lignin input over the pasture), and comparatively higher C/V and medium S/V ratios for LX (tussock grassland). This is in line with the source specific ratios established by Hedges and Mann (1979). The S/V and C/V ratios of all sample types normalized to the respective ratios of Hodges Creek Cave (HC) are presented in Figures 6(a) and 6(b), respectively. This presentation shows that in all three sample types (soil, dripwater and flowstone), HC and NB have the

highest S/V and the lowest C/V ratios, while WP has the lowest S/V, and DC the highest C/V ratios. This means that although the absolute ratios increase from soil to dripwater to speleothem and the magnitude of change varies among cave sites, the relative ratios show the same trend in all sample types. This in turn suggests that the LOP signature of the overlying vegetation is at least partly preserved in dripwater and speleothem samples, which is a precondition for using LOPs as a biomarker for past vegetation changes. Possible causes for the different magnitude of change of the LOP ratios from soil to dripwater to flowstone

among the different cave sites can be differences in soil thickness, types of soil and vegetation density, which can all influence transport and degradation of LOPs.

As expected, the LOP concentrations decrease from the soil to the cave. However, the C-, S-, and V-group LOPs do not diminish at the same rate. The decrease is strongest for the V-group and weakest for the C-group (Fig. 3(f)). This apparent

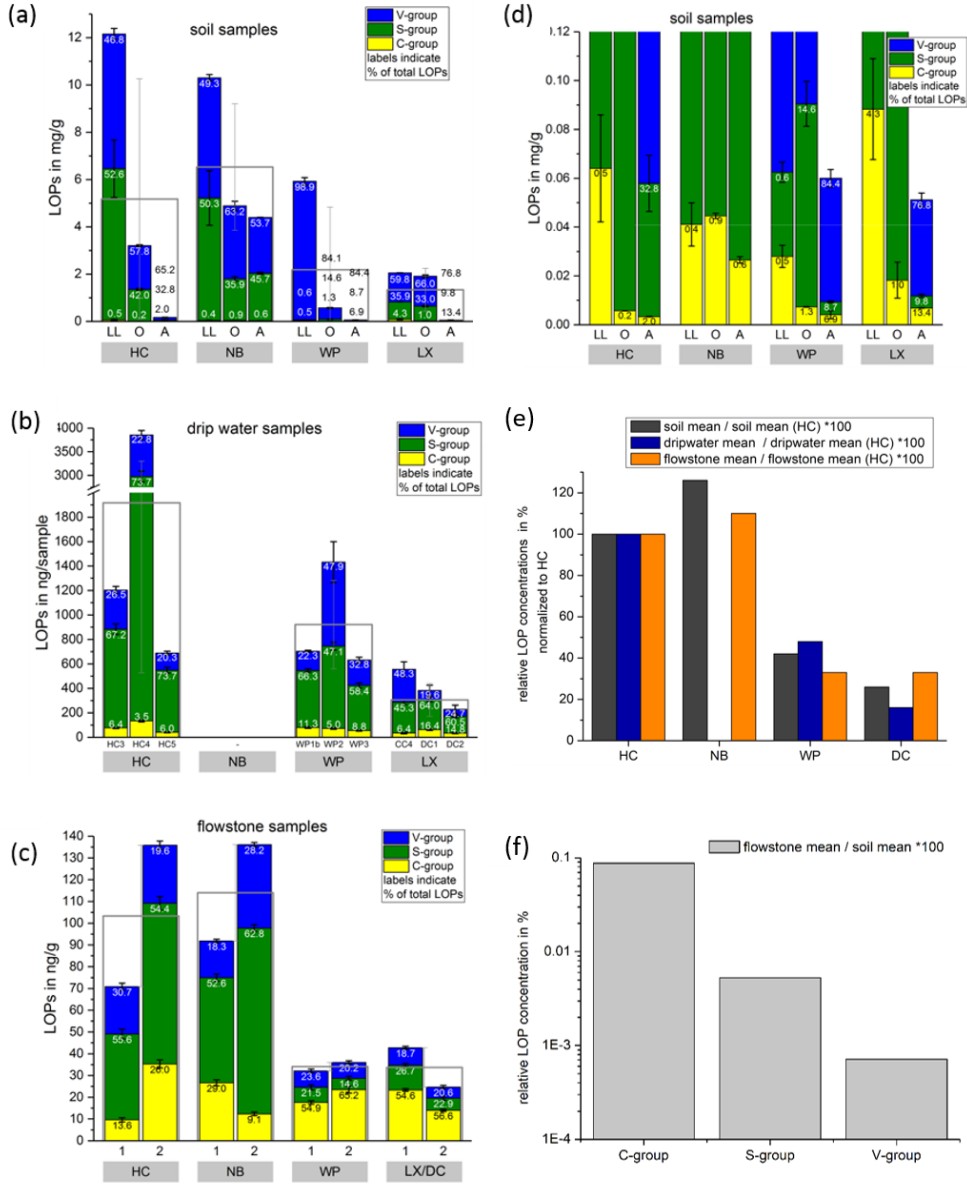

**Figure 3.** Comparison of LOP concentrations in (a) soil, (b) passively-sampled dripwater and (c) flowstone samples. The height of the columns represents the total LOP concentration, Σ8, and the different colors refer to the C-, S-, and V-group LOPs. The empty boxes show the mean values of the Σ8 concentrations. Panel (d) shows an enlargement of panel (a) to better illustrate the concentrations of the C-group LOPs. In panel (e), the mean LOP concentrations (represented as the empty boxes in panels (a), (b) and (c)) of soil, dripwater and flowstone samples are normalized to the respective mean LOP concentration of HC. Panel (f) illustrates the different decreases between the soil and the flowstone of the C-, S- and V-group LOPs. The abbreviations in the grey boxes in panels (a) to (d) indicate the cave (see main text). Other abbreviations: LL = leaf litter, O = organic horizon, A = A-horizon. In panel (b), different bars correspond to the individual drip sites (see main text). In panel (c), sample 1 stands for the most recently deposited flowstone sample from the cave floor (distance from top 0–1 cm), whereas 2 stands for the stratigraphically older flowstone sample with a distance from top of 1–2 cm. The ages of the flowstone samples are given in Table 1.

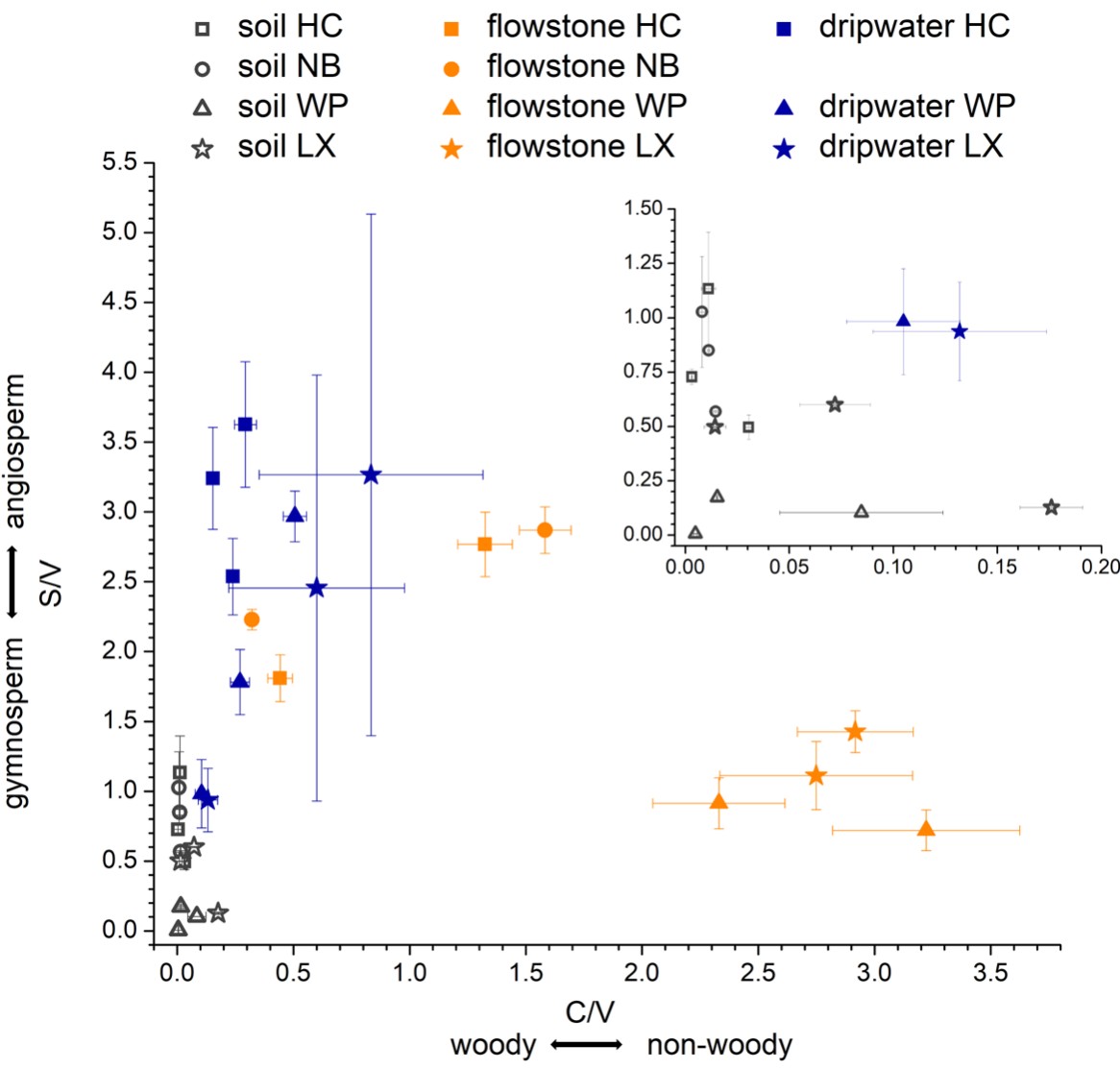

**Figure 4.** S/V versus C/V values of soil, dripwater and flowstone samples from Hodges Creek Cave (HC), Nettlebed Cave (NB), Waipuna Cave (WP) and Luxmore Caves (LX). The insert shows an enlargement of the area close to the origin.

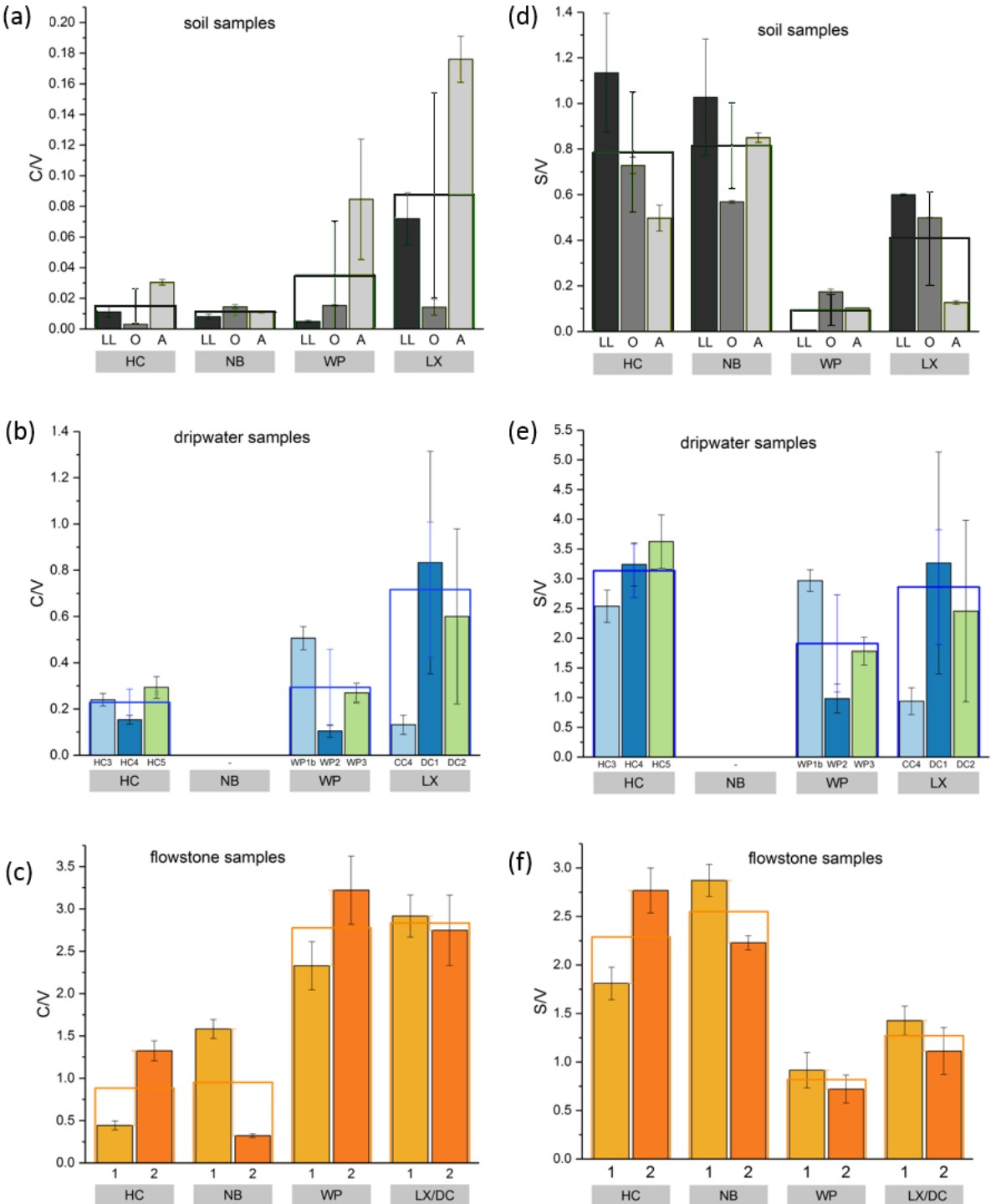

**Figure 5.** C/V and S/V values of the individual soil, dripwater and flowstone samples. The empty boxes show the mean values of the individual samples.

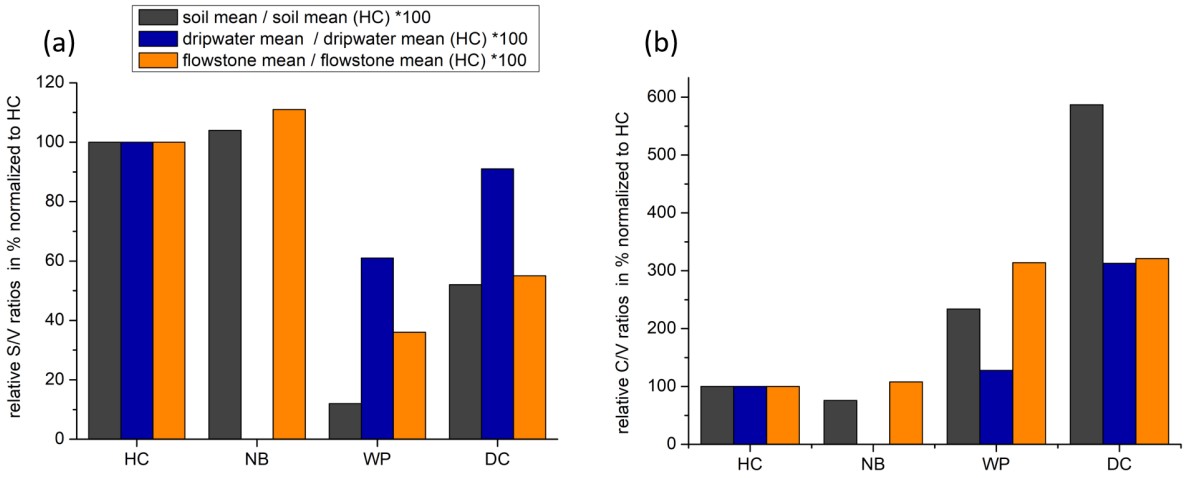

**Figure 6.** Mean S/V ratios (a) and mean C/V ratios (b) of soil, dripwater and flowstone samples normalized to the respective mean S/V or C/V ratios of HC.

change in the lignin composition could be caused by biotic factors, such as differential microbial degradation, or abiotic factors, such as the interaction with mineral surfaces. Degradation studies of plant litter in litter bags or *in situ* in the top 25 cm of the soil using CuO oxidation showed that the concentrations of C- and S-group LOPs decreased faster than the concentrations of V-group LOPs (Bahri et al., 2006; Opsahl and Benner, 1995; Jex et al., 2014). However, the faster degradation of lignin
high in S- and C-group LOPs does not necessarily mean that the released lignin fragments are completely mineralized. It is also possible that they are leached to deeper soil layers since the (partial) degradation leads to a higher oxidation state, i.e., to more carboxylic acid functional groups, and thus to a better water solubility. Concerning the abiotic factors, Hernes et al. (2007) leached different plant litters in water and observed changes in the C/V ratio from -40% to up to +400% in the leachate compared to the parent litter. Subsequent sorption of the leachate to different mineral phases further increased the C/V ratio up
to sevenfold compared to the leachate. For S/V, Hernes et al. (2007) observed a twofold increase by the combined effects of leaching and sorption, and the acid/aldehyde ratios increased as well by a factor of 2 or more. Both the differential microbial degradation and the effects of leaching and sorption are probably caused by structural differences between the individual lignin monomers. Lignin high in sinapyl (S-group) monomers contains mainly $\beta$-aryl ether [$\beta$-O-4] linkages, which can be chemically cleaved more easily, whereas lignin with high proportions of guaiacyl (V-group) monomers contains higher amounts
of phenylcoumaran [$\beta$-5], biphenyl [5-5] and biphenyl ether [5-O-4] linkages, which are more resistant to chemical cleavage (Boerjan et al., 2003). As a result, lignin with a high contribution of S units is more easily degraded and also less branched because the [5-O-4] and [5-5] linkages represent branching points in the polymer. The C-group phenols are mainly bound by peripheral ester-linkages linking lignin and cellulose and are often not part of the lignin polymer itself (Kögel-Knabner, 2002; Boerjan et al., 2003). This part of the C-group phenols can be cleaved from the lignin by mild base hydrolysis (Opsahl and
Benner, 1995). Probably, hydrolysis of the ester-bound C-group LOPs also occurs in the soil and the epikarst aquifer above

the cave. These hydrolyzed C-group phenols are more water soluble and may be transported to the cave more efficiently than the less soluble larger lignin particles and therefore can be enriched in cave dripwater and speleothems compared to S- and V-group phenols. This could also explain the increase of the C/V ratios from the leaf litter (LL) to the O and A horizons for all cave sites in our results (Fig. 5). The S/V ratios, on the other hand, decrease from the leaf litter to the O and A horizons, consistent with a higher degradation state in deeper soils, but are much higher in the dripwater and the flowstone samples than in the soil. A possible explanation lies in the fact that the adsorption of lignin from the dripwater to the XAD material and the subsequent elution as well as the incorporation of lignin into the speleothem fabric present phase change processes as well, which can be selective and cause fractionation.

Our results suggest that the lignin fragments high in C- and S-group lignins, which are more easily degraded, are preferentially transported to deeper soils, probably due to their better water solubility. Thus, they are enriched in dripwater and speleothems compared to the more stable V-group lignins, which are possibly preferentially retained by sorption to mineral surfaces. However, our results also show that the initial signals from the vegetation sources, i.e., the relative differences in $\Sigma 8$, C/V and S/V between the different cave sites, are well preserved in the dripwater and flowstone samples despite of these processes.

### 3.2 Dripwater study from Waipuna Cave

Figure 7 shows the results of the active dripwater sampling at Waipuna Cave. The total LOP concentrations, $\Sigma 8$, in the filtered dripwater samples were very low, with a range of 2–290 $\mathrm{ng \cdot L^{-1}}$. In comparison, $\Sigma 8$ in the dripwater from the German Herbstlabyrinth cave system was in the range of 500–1800 $\mathrm{ng \cdot L^{-1}}$ in summer and 30–400 $\mathrm{ng \cdot L^{-1}}$ in winter (Heidke et al., 2019). The analytes of the S-group were below the detection limit in all samples, and the analytes of the C-group were below the detection limit in most samples. Therefore, the C/V and S/V ratios could not be calculated. We suspect that the lignin was not dissolved homogeneously in the water samples, but instead adsorbed to particulate matter. In this case, the filtering of the samples would have affected the LOP concentrations. Therefore, we analyzed the particulate matter retained in the syringe filters. The results are shown as the hatched bars in Fig. 7. The two samples that had a considerable particulate matter load, WP2-04 and WP2-07, showed the highest LOP concentrations of about 2200 $\mathrm{ng \cdot sample^{-1}}$. WP2-11 and WP-stream had low LOP concentrations of about 170 and 50 $\mathrm{ng \cdot sample^{-1}}$, whereas the rest of the samples was close to or below the detection limit.

The Waipuna LOP results thus suggest that lignin in dripwater is at least partly transported by particles or colloids with dimensions $> 0.45\,\mu\mathrm{m}$, which will be favored by flow along fractures rather than seepage flow (which can allow the migration of low molar mass organic acids and nanoparticles; Hartland et al. (2011, 2012)). Lignin is likely to be adsorbed to mineral particles (Theng, 2012) and can also form organic colloids by coiling with other organic substances (humic matter). The adsorption and desorption of LOPs or lignin to mineral surfaces is a continuous and repetitive process that depends on many parameters, as described in the soil continuum model by Lehmann and Kleber (2015), the regional chromatography model by Shen et al. (2015), and observed for example in the leaching and sorption experiments by Hernes et al. (2007). Our results show that the particles collected in the filters can be (at least partially) desorbed and eluted with methanol. In previous studies

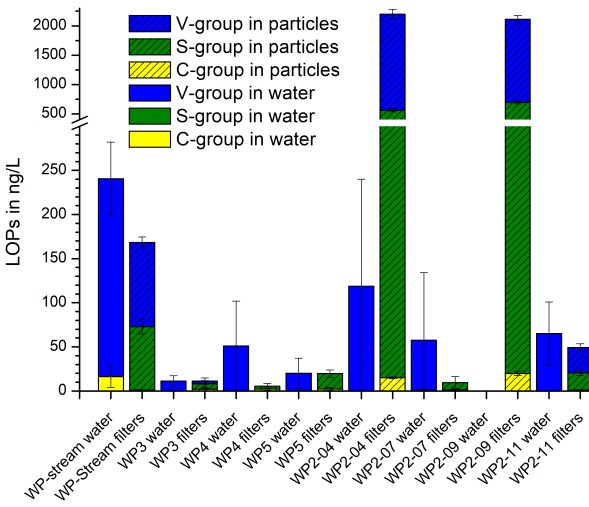

**Figure 7.** LOP concentrations of the filtered dripwater samples from Waipuna Cave and of the particulate matter retained in the filters.

of lignin oxidation products in cave dripwater and speleothems, the samples were either not filtered before SPE extraction, or 1.0 μm glass fiber filters were used, which possibly accounts for the large difference between the LOP results of filtered dripwaters from Waipuna and non-filtered dripwaters from the Herbstlabyrinth (Heidke et al., 2018, 2019). These findings should be systematically investigated in future to better understand the transport processes of lignin into the cave system and to find a more suitable sampling method for lignin analysis in dripwater. For example, passive sampling methods with adsorption resins or filters could be more efficient and also easier to apply than whole water sampling. The contribution to the speleothem record of lignin transported by particles compared to lignin transported in solution and how they differ in composition is difficult to estimate on the basis of the current data. To control and reduce the number of influencing parameters, we suggest using a combination of cave monitoring projects and artificial cave setups to study this question.

## 4   Conclusions

Our results demonstrate that, from the soil to the flowstone, the C/V and S/V ratios both increase while the total lignin content, $\Sigma 8$, strongly decreases. Nevertheless, the relative LOP signal from the overlying soil at the different cave sites is preserved in the flowstone. This shows that the LOP signal in speleothems can be used as a proxy for the lignin input in the overlying soil and therefore as a proxy for past vegetation changes. However, for the interpretation of C/V and S/V ratios, it is important that only samples of the same type (e.g., speleothem, dripwater or soil) are compared and only relative variations are examined, since the LOP signal is strongly influenced by transport and degradation processes. We suggest that a faster oxidative degradation of C- and S-group LOPs in the soil compared to the more stable V-group LOPs makes them more soluble, which in turn leads to a more efficient transport of these LOPs from the soil to the cave causing higher C/V and S/V ratios in the speleothems. On the

other hand, LOPs can also adsorb to mineral particles and be transported to the cave via particle transport, which may partially override the aforementioned fractionation.

For the interpretation of paleo-vegetation records from speleothems using LOPs, it is advisable to first analyze some recently deposited speleothem samples and compare these with samples from the overlying soil to 'ground-truth' the LOP signals for the specific cave site. However, it is not clear yet whether the relationship between the LOP signals in the soil and those in the flowstones changes over time, for example with changing soil thickness and climatic conditions. The transformation of LOP signals from the soil to the cave should be further investigated by expanding the sample set to a greater number of different cave sites with different vegetation, soil type and thickness and climatic conditions to identify and quantify different influences. In addition, the incorporation of lignin into the speleothem fabric could be studied using artificial cave setups (Hansen et al., 2017, 2019; Polag et al., 2010; Wiedner et al., 2008).

*Data availability.* All relevant data is included in the manuscript and the supplement.

*Author contributions.* Conception and design of the work was done by IH, AH, DS and TH. Field work and sample collection was done by AH, JH, AP and IH. Data collection, performing of the experiments and drafting of the article was done by IH. Data analysis and interpretation, critical revision of the article and final approval of the version to be published was done by all authors.

*Competing interests.* The authors declare that they do not have any competing interests.

*Acknowledgements.* Denis Scholz acknowledges funding from the German Research Foundation (SCHO 1274/9-1 and SCHO 1274/11-1). This study received funding from the European Union's Horizon 2020 Research and Innovation programme under the Marie Skłodowska-Curie grant agreement No 691037, Marsden Fund grant (UOW1403), and Rutherford Discovery Fellowship (RDF-UOW1601) to Adam Hartland.

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
