# Peer review of "Lignin oxidation products in soil, dripwater and speleothems from four different sites in New Zealand"

_Biogeosciences, 2020_

## Referee Comment (RC1) · Anonymous Referee #1 · 27 Nov 2020

This paper presents a quite interesting dataset comparing lignin phenol products (LOPs) in the dripwater, speleothems and the overlaying soil layers at four different sites in New Zealand, aiming to test if LOPs in speleothems may loyally record the local vegetation information. There is so far very limited information on LOPs in speleothems and dripwater and this study is a nice complement to related studies published by the authors recently. The data and information carried herein are important for future potential applications of LOPs in speleothems for paleo-reconstruction. However, I think some of the conclusions are too strong and the authors need to further address several concerns to have a robust conclusion.

[Figure]

1. As briefly acknowledged in the paper, the resin method of collecting dripwater organic matter needs to be tested for (i) potential fractionation of lignin monomers; (ii) loading capacity of DOC; (iii) effects of potential degradation for the one-year sampling time.

2. Although particulate lignin may not be completely solubilized and analyzed in the study, the authors should try to estimate how much particulate lignin may contribute to the record in speleothem (ratio of dissolved versus particulate lignin and how different are they in composition).

3. Page 11, Line 4: I don't quite understand this sentence "the relation between the different cave sites is similar". I do see variations in the magnitude of change from soil to dripwater to speleothem among sites. What is the likely cause? Variability is also high for the same site. This needs to be considered. The conclusion is too strong.

4. Same page, Line 18: Microbial degradation often leads to decreasing C/V and S/V ratios. Hence, it can be ruled out. Also, the C/V ratio increases from LL to A horizon in Fig. 5a. Why? Do you think that plant roots may contain more C (it does happen for some species)?

5. Have you considered to analyze LOPs in soil solution (or WEOM from soils)? They seem to be more appropriate or relevant for dripwater comparison than soils.

6. Conclusions: How would mineralogy of bedrocks affect the sorption and change of LOPs along their vertical transport?

---

## Referee Comment (RC2) · Anonymous Referee #2 · 30 Nov 2020

The manuscript by Heidke et al. deals with the origin, transport and potential use of lignin oxidation products (LOPs) in speleothems as a paleoenvironmental proxy. The authors performed a thoughtful study of LOPs in soils (at different depths), dripwater and carbonate speleothems from several caves of New Zealand. The methodological approach seems reasonable, although I agree with the authors (page 15, lines 5 -10) that the sampling/elution method should be optimized for future studies. The role of filtering should be tested, as well as the principles of organic matter incorporation into calcite need to be evaluated in laboratory. The manuscript is well written and will be of interest for a specialized audience.

[Figure]

I have a general comment about the origin of organics in speleothems. The authors consider that all the LOPs in the speleothems comes from the soils over the studied caves. However, how other sources of organic matter could impact the concentration/nature of organics in speleothems? This may include compounds of fossil origin, like coal strata in the bedrock (e.g. Gázquez et al., 2012) and organic matter introduced by animals (bat guano). If possible, the authors should discuss whether these sources (or others) could have an impact on the characteristics of LOPs in speleothems.

References: Gázquez F., Calaforra J.-M., Rull F., Forti P. and García-Casco A. 2012. Organic matter of fossil origin in the amberine speleothems from El Soplao Cave (Cantabria, Northern Spain). International Journal of Speleology, 41(1), 113-123. Tampa, FL (USA). ISSN 0392- 6672. DOI: http://dx.doi.org/10.5038/1827-806X.41.1.12

---

## Author Comment (AC1) · 9 Jan 2021

Please see supplement.

Please also note the supplement to this comment:
https://bg.copernicus.org/preprints/bg-2020-345/bg-2020-345-AC1-supplement.pdf

---

## Author Comment (AC2) · 9 Jan 2021

Please see supplement.

Please also note the supplement to this comment:
https://bg.copernicus.org/preprints/bg-2020-345/bg-2020-345-AC2-supplement.pdf

---

## Author Response (AR1)

**Response to the referees**

We thank the two reviewers for carefully evaluating our manuscript. Our point-by-point reply directly follows the referee comments (blue) and appears in black after each comment. New text passages in the manuscript are written in green, deleted passages in .

**Response to referee no. 1**

**Comment 1:**

As briefly acknowledged in the paper, the resin method of collecting dripwater organic matter needs to be tested for (i) potential fractionation of lignin monomers; (ii) loading capacity of DOC; (iii) effects of potential degradation for the one-year sampling time.

We agree with the referee that these are important tests for future studies of dripwater organic matter collected on resins. However, we think that our current results can be used without these tests because all resin columns were treated in the same way concerning the type and amount of resin, the elution procedure and the sampling time, and because we compared only relative differences between the different cave sites and between soil, dripwater and speleothem samples and not absolute LOP concentrations or ratios.

**Comment 2:**

Although particulate lignin may not be completely solubilized and analyzed in the study, the authors should try to estimate how much particulate lignin may contribute to the record in speleothem (ratio of dissolved versus particulate lignin and how different are they in composition).

To clarify the terminology: When we talk about lignin transported via particles in comparison to lignin transported in solution, we mainly mean LOPs (or fragments of lignin) that are adsorbed to the surface of mineral particles, and not necessarily particles that consist solely of lignin. The adsorption and desorption of LOPs or lignin to mineral surfaces is a continuous and repetitive process that depends on many parameters, as described in the soil continuum model by Lehmann et al. (2015), the regional chromatography model by Shen et al. (2015), and observed for example in the leaching and sorption experiments by Hernes et al. (2007). Thus, it is hardly possible to estimate the contribution to the speleothem record of lignin transported by particles compared to lignin transported in solution and how they differ in composition, at least on the basis of the current data. However, we agree that this is an interesting and important research question. To control and reduce the number of influencing parameters, we suggest using a combination of cave monitoring projects and artificial cave setups to study this question.

We have amended the text in the manuscript on p. 13, lines 1-11, accordingly:

"The Waipuna LOP results thus suggest that lignin in dripwater is at least partly transported by particles or colloids with dimensions > 0.45 µm, which will be favored by flow along fractures rather than seepage flow (which can allow the migration of low molar mass organic acids and nanoparticles; (Hartland et al. (2011), Hartland et al. (2012)). Lignin is likely to be adsorbed to mineral particles (Theng et al. (2012)) and can also form organic colloids by coiling with other organic substances (humic matter). The adsorption and desorption of LOPs or lignin to mineral surfaces is a continuous and repetitive process that depends on many parameters, as described in the soil continuum model by Lehmann and Kleber (2015), the regional chromatography model by Shen et al. (2015), and observed for example in the leaching and sorption experiments by Hernes et al. (2007). Our results show that the particles collected in the filters can be (at least partially) desorbed and eluted with methanol. In previous studies of lignin oxidation products in cave dripwater and speleothems, the samples were either not filtered before SPE extraction, or 1.0 µm glass fiber filters were used, which possibly accounts for the large difference between the LOP results of filtered dripwaters from Waipuna and non-filtered dripwaters from the Herbstlabyrinth (Heidke et al. (2018), Heidke et al. (2019)). These findings should be systematically investigated in future to better understand the transport processes of lignin into the cave system and to find a more suitable sampling method for lignin analysis in dripwater. For example, passive sampling methods with adsorption resins or filters could be more efficient and also easier to apply than whole water sampling. The contribution to the speleothem record of lignin transported by particles compared to lignin transported in solution and how they differ in composition is difficult to estimate on the basis of the current data. To control and reduce the number of influencing parameters, we suggest using a combination of cave monitoring projects and artificial cave setups to study this question."

Comment 3:

Page 11, Line 4: I don't quite understand this sentence "the relation between the different cave sites is similar". I do see variations in the magnitude of change from soil to dripwater to speleothem among sites. What is the likely cause? Variability is also high for the same site. This needs to be considered. The conclusion is too strong.

To better explain the meaning of the sentence "the relation [of the C/V and S/V ratios] between the different cave sites is similar", we created another diagram similar to Figure 3(e) (new Figure 6 in the revised manuscript). In this figure, the ratios S/V (Figure 6 (a)) and C/V (Figure 6 (b)) are normalized to the respective ratios of Hodges Creek Cave (HC). This presentation shows that in all three sample types (soil, dripwater and flowstone), HC and NB have the highest S/V and the lowest C/V ratios, while WP has the lowest S/V, and DC the highest C/V ratios. This means that although the magnitude of change in LOP ratios from soil to dripwater to speleothem indeed varies among cave sites, the relative ratios show the same trend in all sample types. This in turn suggests that the LOP signature

of the overlaying soil is at least partly preserved in dripwater and speleothem samples.

[Figure]

Figure 6:  Mean S/V ratios (a) and mean C/V ratios (b) of soil, dripwater and flowstone samples normalized to the respective mean S/V or C/V ratios of HC.

Possible causes for the different magnitude of change of the LOP ratios from soil to dripwater to flowstone among the different cave sites can be differences in soil thickness, types of soil and vegetation density, which can all influence transport and degradation of LOPs.

We have amended the text in the manuscript on p. 10, line 5 to p. 11, line 6 accordingly:

"The C/V and S/V ratios in the leaf litter samples are representative of the current vegetation cover (Fig. 5), with low C/V and high S/V ratios for HC and NB (angiosperm southern beech forest), low C/V and low S/V ratios for WP (gymnosperm podocarp forest, apparently dominating the lignin input over the pasture), and comparatively higher C/V and medium S/V ratios for LX (tussock grassland). This is in line with the source specific ratios established by Hedges and Mann (1979).  Figure 6 shows the S/V (Fig. 6 (a)) and C/V (Fig. 6 (b)) ratios of all sample types normalized to the respective ratios of Hodges Creek Cave (HC). This presentation shows that in all three sample types (soil, dripwater and flowstone), HC and NB have the highest S/V and the lowest C/V ratios, while WP has the lowest S/V, and DC the highest C/V ratios. This means that although the absolute ratios increase from soil to dripwater  to speleothem and the magnitude of change varies among cave sites , the relative ratios show the same trend in all sample types. This  in turn suggests that the  LOP signature of the overlying vegetation  is at least partly preserved in dripwater and speleothem samples, which is a precondition for using LOPs as a biomarker for past vegetation changes. Possible causes for the different magnitude of change of the LOP ratios from soil to dripwater to flowstone among the different cave sites can be differences in soil thickness, types of soil, and vegetation density, which can all influence transport and degradation of LOPs."

**Comment 4:**

Same page [p. 11], Line 18: Microbial degradation often leads to decreasing C/V and S/V ratios. Hence, it can be ruled out. Also, the C/V ratio increases from LL to A horizon in Fig. 5a. Why? Do you think that plant roots may contain more C (it does happen for some species)?

Possible reasons for the increase of C/V ratios from LL to A horizon and the increase of C/V and S/V ratios from soil to dripwater and flowstones have been discussed in detail on p. 11, lines 6 to 35. In brief, the main reasons probably are the following: First, the combined effects of leaching and sorption as described by Hernes et al. (2007) (discussed in lines 14-18). Second, the higher oxidation state (more carboxylic acid functional groups) after degradation of C- and S-group LOPs, which leads to better water solubility and thus improved transport of C- and S-group LOPs compared to the less degraded V-group LOPs (lines 11-14). And third, the hydrolysis of the ester-bound C-group LOPs (lines 24-30). We do not think that the higher C content of plant roots plays an important role here, since the main contribution to the lignin pool in soils probably is from rotting leave litter and wood rather than plant roots. However, we cannot rule out a partial contribution of roots.

**Comment 5:**

Have you considered to analyze LOPs in soil solution (or WEOM from soils)? They seem to be more appropriate or relevant for dripwater comparison than soils.

We agree that the water extractable organic matter (WEOM) might be more relevant for comparison with dripwater than the organic matter that can be extracted from soils via CuO oxidation with NaOH at elevated temperature. This might lead to an overdetermination of LOPs in soils, but the relative changes should still be similar, and the conclusion would not be changed. We used this method for better comparison of our soil and leaf litter results with published LOP ratios for different vegetation types, which used the same methodology (e.g., Hedges and Mann 1979). However, we will consider using WEOM instead of, or in addition to, soil samples in future studies.

**Comment 6:**

How would mineralogy of bedrocks affect the sorption and change of LOPs along their vertical transport?

As the laboratory studies of Hernes et al. (2007) and Hernes at al. (2013) show, different soils and minerals have a significant influence on the sorption and fractionation dynamics of LOPs, which also depend on the lignin monomers, the conformation of lignin fragments and many other parameters. This was shortly discussed on p. 11, lines 14-18. However, it was not the goal of our present study to elucidate these influences, as this can be better done in laboratory studies with controlled parameters, such as artificial cave chambers.

**Response to referee no. 2**

**Comment 1:**

I have a general comment about the origin of organics in speleothems. The authors consider that all the LOPs in the speleothems comes from the soils over the studied caves. However, how other sources of organic matter could impact the concentration/nature of organics in speleothems? This may include compounds of fossil origin, like coal strata in the bedrock (e.g. Gázquez et al., 2012) and organic matter introduced by animals (bat guano). If possible, the authors should discuss whether these sources (or others) could have an impact on the characteristics of LOPs in speleothems.

It seems plausible that LOPs can also originate from coal strata in the bedrock, at least from low-grade coal. However, to our knowledge, there are no coal strata present in the bedrock of the studied cave sites. The flowstone samples were of rather light color, no dark amberine colors as in Gazquez et al. (2012) were observed. Only NB and HC showed some darker colored layers, but these could also originate from soil organic matter or trace elements.

Bat guano probably does not play a significant role in flowstone LOPs from the studied cave sites. There are two native bat species in New Zealand. The long-tailed bat is an aerial insectivore, therefore its guano should not contain significant amounts of plant material. The short-tailed bat also feeds on fruit, nectar, and pollen in addition to insects, thus its guano could possibly contain traces of lignin. However, no visible bat guano layers were found on the flowstone surfaces, and the outer layer of calcite was edged away before the samples were dissolved to remove debris from the surface. Nevertheless, we cannot completely rule out a contribution of bat guano to the older flowstone samples.

**References**

Gázquez F, Calaforra J-M, Rull F, Forti P, García-Casco A (2012) Organic matter of fossil origin in the amberine speleothems from El Soplao Cave (Cantabria, Northern Spain). IJS 41(1):113–123

Hedges JI, Mann DC (1979) The characterization of plant tissues by their lignin oxidation products. Geochimica et Cosmochimica Acta 43(11):1803–1807

Hernes PJ, Robinson AC, Aufdenkampe AK (2007) Fractionation of lignin during leaching and sorption and implications for organic matter "freshness". Geophys. Res. Lett. 34(17):1921

Hernes PJ, Kaiser K, Dyda RY, Cerli C (2013) Molecular trickery in soil organic matter: hidden lignin. Environmental science & technology 47(16):9077-9085

Lehmann J, Kleber M (2015) The contentious nature of soil organic matter. Nature 528(7580):60

Shen Y, Chapelle FH, Strom EW, Benner R (2015) Origins and bioavailability of dissolved organic matter in groundwater. Biogeochemistry 122(1):61–78

---

## Author Response (AR2)

Dear editor,

Please find our replies to your comments directly under each comment.

Best regards,

Inken Heidke

Dear authors,

Your revised manuscript has been sent out to the original two reviewers, whose feedback was invaluable in the improvement of your manuscript. One of the reviewers provided the following comment: "The authors' replies to my previous comments are mostly adequate. However, a caveat on the resin method should be added into the text: although comparative differences may be used to indicate the relative abundance of organic matter captured by the same method and procedure, overloading (if present) may obscure actual (presumably larger) variations between samples. Hence, the authors should at least provide considerations or rationale to rule out this possibility." Can you please address this?

To address this, we included the following paragraph in the introduction of the paper:

"The vast majority of procedures for the determination of organic analytes in aqueous solutions such as drip water rely on the collection of discrete samples of the water at a specific time. Subsequent laboratory analysis of the sample then provides a snapshot of the concentration of the target analytes at the time of sampling. In the presence of fluctuating concentrations, this method also has drawbacks, such as allowing episodic concentration fluctuations to be missed. One solution to this problem is to increase the frequency of sampling or to install automated sampling systems that can collect numerous drip water samples over a period of time. However, this is costly and in many cases impractical. But alternatives exist to overcome some of these difficulties. Of these, passive sampling methods have shown to be promising tools for measuring aqueous concentrations of a wide range of organic substances. Passive samplers avoid many of the problems described above because they enrich target analytes in situ, can be used for extended periods of time, and can be applied without continuously accessing the sampled caves. The goal is to determine the mass of target components accumulated by a sampler, thereby obtaining time-averaged concentrations. Of course, passive samplers also have their disadvantages (Vrana et al., 2005), which is why they are used here as one method in combination with other objects of investigation (leaf litter, soil as well as flowstone samples) and are primarily used to compare the different sampled sites with each other."

As an Editor, I also found several suggestions for improvements:

Section 2.1.: Please include specific lat/long for each cave.

Done, in both the manuscript and the supplement.

Table 1: the dft in the table is quite confusing. Please indicate with a dot (or a relevant shape) in your supplementary figure, where exactly the trenches were drilled.

Since the dating samples were drilled from a different slab of the flowstones than the lignin samples, we have included photographs of the slabs with the drilled trenches in the supplementary information. Both slabs have a scale to their site for comparison. In addition, the table with the dft of the lignin samples was revised to make it clearer.

Also, in the supplementary, remove the text " S3 Photographs of the flowstone samples", it is confusing as it represents neither a caption nor a sentence.

This is the heading of section S3 "Photographs of the flowstone samples", analogous to section S1 "Analytical methods", section S2 "Description of the cave sites", and section S4 "230 Th/U-dating of flowstone cores". The layout is the default layout of Copernicus. Do you want me to remove all section headings or change just this heading?

Figure 4: Please use a real arrow rather than < and > to indicate the end-members.

Done.

On p.10 of your revised manuscript, please rephrase the newly added text: The S/V and C/V ratios of all sample types normalized to the respective ratios of Hodges Creek Cave (HC) are presented in Figures 6a and 6b, respectively.

Done.

Finally, I'd like the authors also to carefully consider and revise the content of the supplementary materials. There are a lot of missing information, marked with "?". In fact, some of these information can be easily found in the manuscript, and putting them in the supplementary is redundant.

The question marks were placeholders for references in the LaTeX code, I am sorry for having overseen these. They have now been removed and replaced by the appropriate references.

The authors made a statement about the data availability that "All relevant data is included in the manuscript and the supplement." However, only the U-Th data and the vegetation/soil above the cave are presented. Please include also the table data used to make Figures 3–7. These are the core of the manuscript, however, these data are missing. Please fix these.

Done.

Thank you for addressing these comments/suggestions. Looking forward to your revision,

Best wishes,

Dr. Ny Riavo G. Voarintsoa